# The cancer cell proteome and transcriptome predicts sensitivity to targeted and cytotoxic drugs

Mattias Rydenfelt[1],*, Matthew Wongchenko[2],* (iD), Bertram Klinger[1,3] (iD), Yibing Yan[2], Nils Blüthgen[1,3] (iD)

**Tumors of different molecular subtypes can show strongly deviating responses to drug treatment, making stratification of patients based on molecular markers an important part of cancer therapy. Pharmacogenomic studies have led to the discovery of selected genomic markers (e.g., BRAF^{V600E}), whereas transcriptomic and proteomic markers so far have been largely absent in clinical use, thus constituting a potentially valuable resource for further substratification of patients. To systematically assess the explanatory power of different -omics data types, we assembled a panel of 49 melanoma cell lines, including genomic, transcriptomic, proteomic, and pharmacological data, showing that drug sensitivity models trained on transcriptomic or proteomic data outperform genomic-based models for most drugs. These results were confirmed in eight additional tumor types using published datasets. Furthermore, we show that drug sensitivity models can be transferred between tumor types, although after correcting for training sample size, transferred models perform worse than within-tumor–type predictions. Our results suggest that transcriptomic/proteomic signals may be alternative biomarker candidates for the stratification of patients without known genomic markers.**

## Introduction

Current cancer therapies often have low patient benefit-to-risk ratios, where negative side effects might be severe even when efficacy is only moderate. To determine which patients would (or would not) benefit from a given therapeutic, great efforts have been directed into discovering and validating biomarkers for therapeutic response, often within the mutational landscape of tumors. However, success has been limited to few examples. Cancer cell line panels can be useful in vitro tools to derive relevant biomarkers (Barretina et al, 2012; Garnett et al, 2012; Cook et al, 2014; Costello et al, 2014; Klijn et al, 2014; Aben et al, 2016; Haverty et al, 2016; Li et al, 2017) pertaining to intracellular processes. Through significant cost

reductions in performing high-throughput experiments and advancements in laboratory automation, such panels can now cover hundreds or even thousands of cell lines and drugs (Barretina et al, 2012; Garnett et al, 2012), providing ample data to search for new biomarkers, and also to address mechanistic questions, such as finding the mechanism of drug action or understanding synthetic lethality (Rees et al, 2016; McDonald et al, 2017). However, as these large screens typically pool cell lines from different tumor types, biomarkers often significantly co-occur with specific tumor types. A recent study has shown that such "cross-entity" biomarkers are rarely predictive within a panel of cell lines from a single tumor type, but only across different tumor types (Iorio et al, 2016). For example, the BRAF^{V600E/K} mutation is a predictive biomarker for MEK inhibitor sensitivity across multiple tumor types, but not within melanoma cell lines specifically (Iorio et al, 2016), although BRAF^{V600E/K} is predominantly found in melanoma (Hodis et al, 2012). This often renders cross-entity biomarkers too unspecific to be used to stratify patients, as the tissue of origin has similar predictive power.

Tumor cells are products of microevolution by which new capabilities are sequentially acquired through accumulation of both genomic and epigenomic alternations, resulting in aberrant activation of signaling pathways (commonly targeted by novel drugs). As different mutations or epigenetic alterations may result in a similar transcriptomic or proteomic state, we reasoned that these states themselves might be better predictors of drug sensitivity than genomic data. Indeed, in a recent study they predict, and experimentally verify, drug sensitivity of the MEK inhibitor trametinib from proteomic markers in melanoma cell lines (Rožanc et al, 2018). To systematically compare genomic, transcriptomic, and proteomic data as predictor of drug sensitivity for many different drugs within a given tumor type, we collected these data in a large panel of melanoma cell lines. For melanoma, multiple targeted drugs (BRAF/MEK inhibitors) have been approved in recent years and extended survival for patients with BRAF^{V600E/K} mutations. Yet, BRAF mutation status is the only known biomarker of BRAF inhibitor sensitivity and no biomarker exists for BRAF wild-type melanoma patients. Furthermore, even within the BRAF-selected populations,

---

[1]Charité–Universitätsmedizin, Institute of Pathology, Berlin, Germany   [2]Genentech Inc., Oncology Biomarker Development, South San Francisco CA, USA   [3]Humboldt Universität zu Berlin, Integrative Research Institute for the Life Sciences, Berlin, Germany

Correspondence: nils.bluethgen@charite.de; yan.yibing@gene.com
*Mattias Rydenfelt and Matthew Wongchenko co-first author

many patients fail to respond to targeted treatments, suggesting that additional biomarkers could help to further personalize treatment options. Using our dataset, we set out to systematically investigate which data category has the most explanatory power of drug sensitivity and derive predictive within-tumor–type biomarkers using cross-validated machine learning. We also used publically available data from pan-cancer cell line panels to validate our findings.

# Results

## BRAF$^{V600E/K}$ mutation status predicts drug sensitivity of BRAF inhibitors but not of other targeted or cytotoxic drugs

In a panel of 49 melanoma-derived cell lines, we sequenced oncogenes that are commonly mutated (Tsao et al, 2012) in melanoma (BRAF, NRAS, KRAS, see Fig 1A). In agreement with what has been observed in melanoma patients (Hodis et al, 2012), we found that the dominating mutation in this cell line panel was BRAF$^{V600E/K}$ (34 of 49), whereas other mutations in BRAF, NRAS, or KRAS were less common (5/49, 8/49, and 1/49). To study the relationship between biomarkers such as BRAF mutation status and drug sensitivity, the cell line panel was subjected to 109 different drugs, representing a wide range of substances targeting, among others, the MAPK pathway, the PI3K/Akt pathway, as well as mitotic and metabolic processes. Drug sensitivities were summarized as the mean viability across evenly spaced drug concentrations in log-space (illustrated in Fig 1B), or equivalently the area under the response curve (AUC), a measure that has proven more reliable in larger drug screens than IC50 (Haverty et al, 2016), and which does not show saturation effects such as maximum inhibition. Since many of the screened drugs showed similar responses across all cell lines, we selected the 27 drugs with highest coefficient of variation in AUC for further analysis. Normalized AUC data for the selected drugs are shown in Fig 1C (Supplemental Data D5), where corresponding unsupervised hierarchical clustering shows that drugs sharing targets predominantly cluster together. Drug sensitivities correlate strongly for drugs that target the same molecule, moderately for drugs that target the same pathway, and little or not at all for drugs targeting different pathways (examples in Fig 1D).

We divided the cell lines into two groups based on BRAF$^{V600E/K}$ mutation status and compared the AUC distributions for each drug using Welch's $t$ test, corrected for multiple hypothesis testing (Benjamini & Hochberg, 1995) (Figs S1 and 1E inset). For the BRAF inhibitors, vemurafenib and dabrafenib, BRAF$^{V600E/K}$ mutation status was a strong predictor of AUC (adjusted $P$ = 4.2 × 10$^{-7}$ and $P$ = 1.9 × 10$^{-5}$, respectively), as expected. Response to cobimetinib, a drug that targets MEK, a protein kinase immediately downstream of BRAF in the MAPK pathway, showed some correlation to BRAF$^{V600E/K}$ mutation status, albeit much weaker and not significant after multiple hypothesis correction. For responses to other drugs, BRAF$^{V600E/K}$ was unpredictive ($P$ > 0.05). Our results show that the key driver mutation in melanoma, BRAF$^{V600E/K}$, was only a strong predictor of drug sensitivity for drugs that target the mutated molecule itself.

Of the 49 cell lines in the panel, some originated from the same patient and could hence not technically be regarded as independent. However, as is shown in Fig S23, removing these duplicates (nine excluded cell lines), did not affect the main conclusions of Figs 1, 2, 3, and 4.

## Full-exome sequencing state still only predicts drug sensitivity of BRAF inhibitors

Exome sequencing data were acquired for 45 out of 49 cell lines in the panel (Supplemental Data D7). We selected mutations reported in at least three but not more than 42 of the cell lines. After additional filtering (see the Materials and Methods section), a total of 1,716 mutations remained. To find a relationship between mutation pattern and drug sensitivity, we decided to build a regression tree (Breiman et al, 1984) model from the 45 × 1,716 dimensional input space, consisting of only ones ("mutated") and zeros ("not mutated"), to predict AUC, as such a model mimics the nonlinear nature of genetic interactions. For each drug, the agreement between measured and predicted AUC after repeated 10-fold cross-validation was quantified by the fraction of variance explained $R^2 = 1 - \frac{\langle(\hat{a}_i - a_i)^2\rangle}{\langle(a_i - \bar{a})^2\rangle}$, where $a_i$, $\hat{a}_i$ are the measured and predicted drug AUC for cell line $i$, and $\bar{a}$ is the mean drug AUC across all cell lines. The observed $R^2$ values were compared with randomized background distributions, where the AUCs of each drug were randomly shuffled before applying the regression tree model ($N$ = 8,192), thus breaking the underlying relationship between input and output (Fig 1E). Again, only the BRAF$^{V600E/K}$-dependent drugs vemurafenib and dabrafenib could be predicted significantly above background (adjusted $P$ < 3.3 × 10$^{-3}$). These two drugs constituted positive controls, showing that the regression tree model was indeed capable of selecting appropriate markers that predict drug sensitivity from exome sequencing data.

To reduce the number of unpredictive passenger mutations, we tried to limit our list of mutations to those reported in the COSMIC Cancer Gene Census (Forbes et al, 2015); this, however, did not lead to an improvement in drug sensitivity predictions (result not shown). We also tried alternative definitions of "mutation" either at the gene or base pair level, but again without any improvements (Fig S2). Taken together, for 27 drugs in our melanoma cell line panel, genomic biomarkers could only be found for BRAF inhibitors, whose drug sensitivity were predicted from oncogenic BRAF itself.

## Proteomic signaling clusters defined by AKT$^{S473}$/PTEN levels do not predict drug sensitivity in melanoma cell lines

To see if drug sensitivities could be predicted for more drugs using other biological data, we acquired proteomic data (Supplemental Data D3) and characterized it using unsupervised hierarchical clustering (Fig 2A). The clustering revealed two distinct groups of cell lines with either high PTEN+low AKT$^{S473}$ or vice versa. Principal component analysis showed that these two groups indeed captured the dominant source of variance across the cell line panel (Fig 2B and C). The switch-like behaviour between PTEN/AKT$^{S473}$ emphasizes the role of PTEN as a strong AKT pathway inhibitor (Maehama & Dixon, 1998; Georgescu, 2010) (Fig 2D). To test whether there was a difference in drug AUCs between the two groups of cell

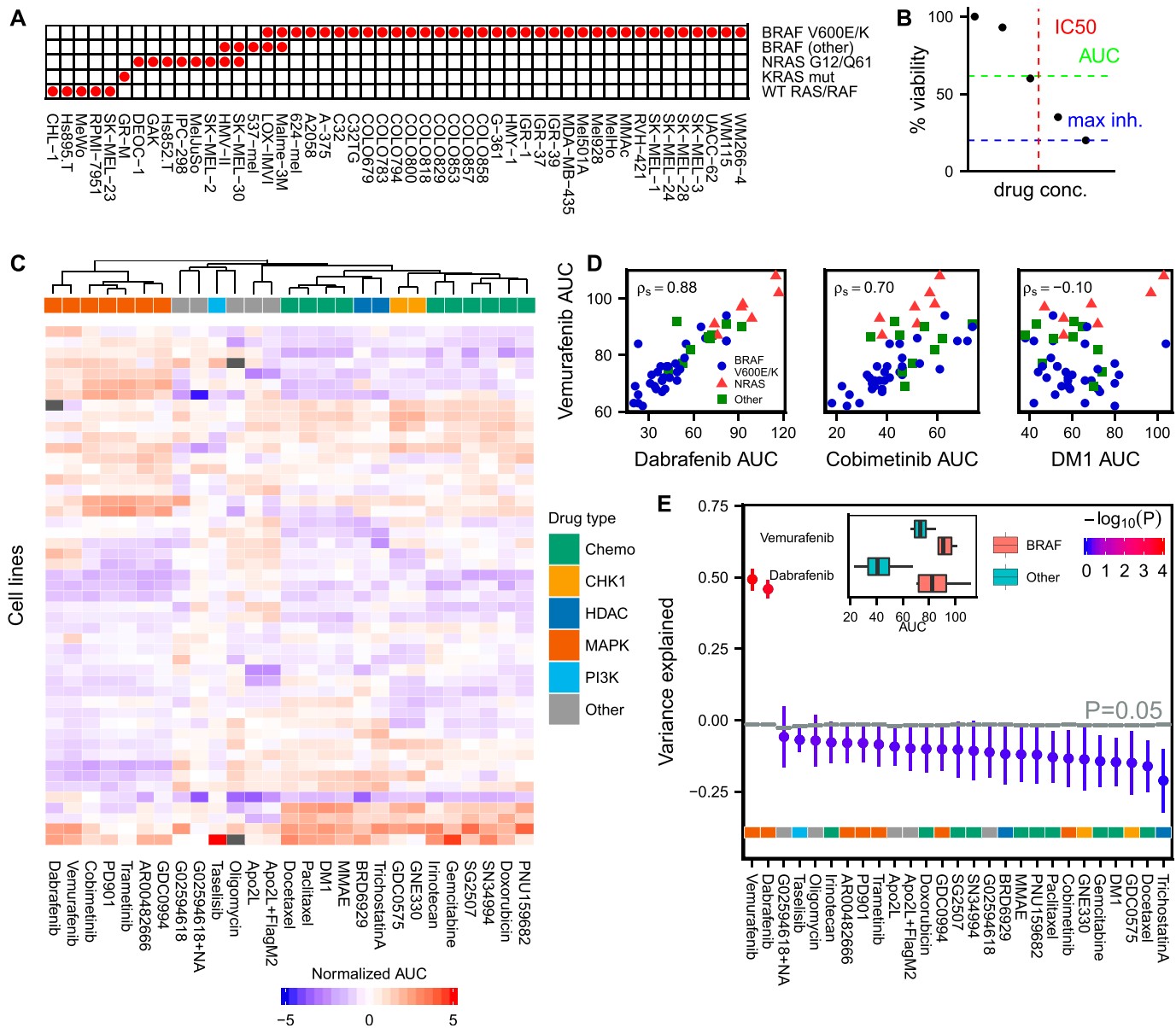

**Figure 1. Predicting drug sensitivity from mutation status.**
**(A)** Melanoma cell line panel with mutation status for commonly mutated oncogenes (BRAF, NRAS, and KRAS). **(B)** Three different drug sensitivity measures defined from a dose–response curve. Throughout this work, we use mean viability, defined as the mean response across evenly spaced drug concentrations in log-space, or equivalently the area under the drug response curve (AUC), as primary proxy of drug sensitivity. **(C)** Normalized drug AUCs for drugs with highest coefficient of variation across the cell line panel (missing values gray). Unsupervised clustering shows that drugs targeting the same molecule or pathway predominantly cluster together. **(D)** AUCs for the two BRAF inhibitors vemurafenib and dabrafenib are strongly correlated and separate cell lines based on BRAF$^{V600E/K}$ mutation status (blue circles). AUCs for vemurafenib and cobimetinib, targeting different molecules in the MAPK pathway, correlate weaker but still substantially. AUCs for vemurafenib and the cytotoxic drug DM1 show no correlation. **(E)** Drug sensitivity predictions from exome sequencing data using a regression tree model. Performance is quantified by fraction of variance explained by the model after repeated 10-fold cross-validation and compared with a background distribution, where AUCs for each drug are randomly shuffled ($N = 8,192$) before running the regression tree algorithm. Gray lines indicate upper 95th percentile of the background distributions. Inset: Cell lines with/without BRAF$^{V600E/K}$ mutation show, as expected, differential response after vemurafenib or dabrafenib treatment. Unless stated otherwise, box plot whiskers define 10th and 90th percentile, and error bars define SD with respect to repeated 10-fold cross-validation throughout the article.

lines, we compared the AUC distributions for each drug using Welch's $t$ test, but none of the 27 tested drugs showed a differential response between the AKT$^{S473}$/PTEN clusters (Fig 2E). However, when we tested the AKT inhibitor ipatasertib (GDC-0068), a drug which had a low coefficient of variation across the cell lines, we

found that it inhibited the growth of cell lines in the AKT$^{S473}$ high cluster more effectively than in the PTEN high cluster (adjusted $P = 3.8 \times 10^{-3}$). This is consistent with previous reports demonstrating increased sensitivity to ipatasertib in cell lines with an activated PI3K/AKT pathway (Lin et al, 2013). Despite having only a modest

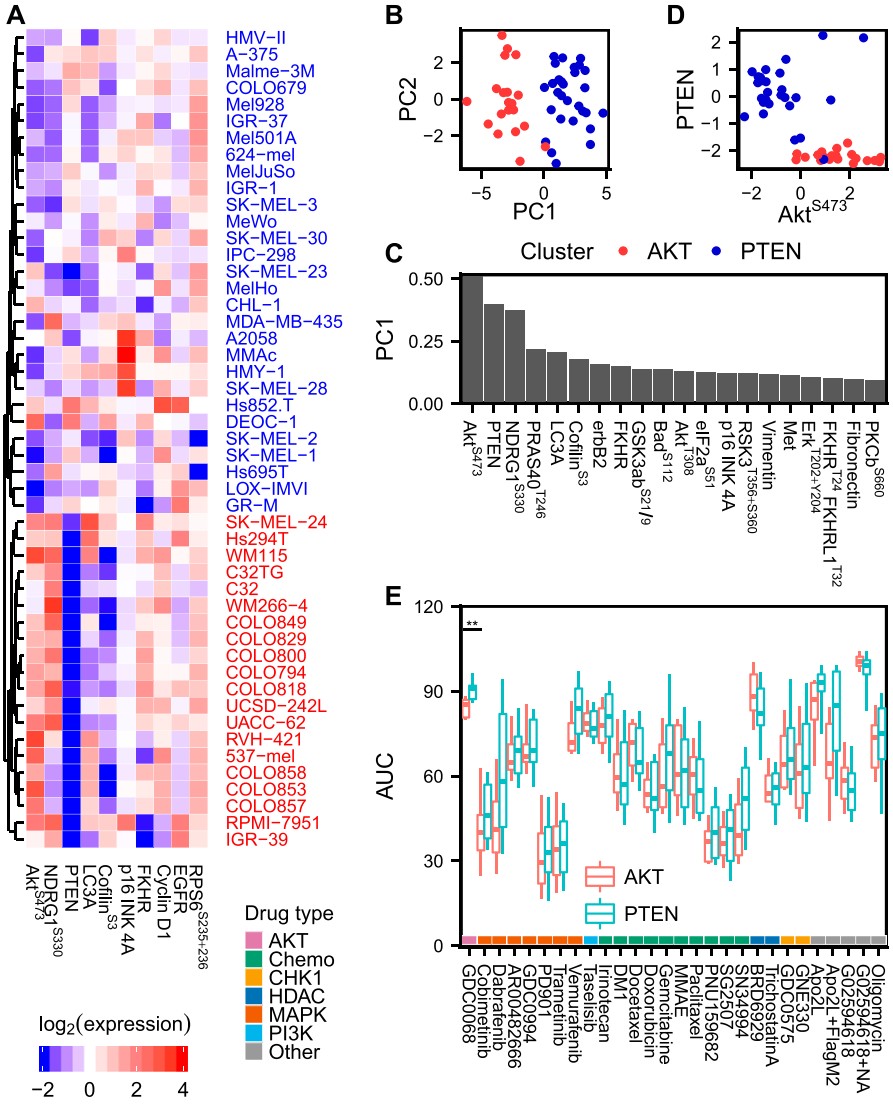

**Figure 2. Predicting drug sensitivity from proteomic (RPPA) data clusters.**
**(A)** Proteomic data of the 10 most variable (phospho) proteins across the cell line panel. Unsupervised hierarchical clustering reveals two distinct groups of cell lines with either high PTEN and low AKT$^{S473}$ or vice versa. **(B, C)** Principal component analysis of proteomic data. The two AKT$^{S473}$/PTEN clusters separate in the first principal component, which is dominated by AKT pathway signals. **(D)** Mutually exclusive expression of PTEN and AKT$^{S473}$. **(E)** Drug AUCs shown separately for cell lines in the AKT$^{S473}$/PTEN clusters. Only the AKT inhibitor ipatasertib (GDC-0068), which was included as a control, showed a differential response between the two clusters.

effect on drug sensitivity in vitro, PTEN loss could still be clinically relevant, for example, by modulating the rate of metastatic outgrowth (Dankort et al, 2009; Gonzalez-Angulo et al, 2011; Wikman et al, 2012; Zhang et al, 2015).

## Proteomic state predicts drug sensitivity for 12 of 27 drugs

The lack of correlation between AKT$^{S473}$/PTEN proteomic clusters and drug sensitivity suggests that a more refined modeling effort to uncover signaling states predictive of drug sensitivity is necessary. Constructing a predictive model directly from the (phospho)proteins (88 antibodies) to drug AUC (49 cell lines), however, presents a challenge because the number of "observations" is smaller than the number of predictors, of which some might be correlated. Partial least squares (PLS) is a statistical method (de Jong, 1993; Wold et al, 2001; Pérez-Enciso & Tenenhaus, 2003; Boulesteix & Strimmer, 2007; Fallahi-Sichani et al, 2015) designed for these situations and reduces the risk of overfitting by dimensionality

reduction of the input space, such that the new dimensions are maximally correlated with the output (Fig 3A and see the Materials and Methods section). By increasing the number of PLS components, defined as linear combinations of the original variables, one can successively reduce the model (training) error.

A separate PLS model was trained for each drug across cell lines and the agreement between measured and predicted AUC after repeated 10-fold cross-validation was quantified using Spearman rank correlation (examples in Fig 3B).

The observed correlations were compared with random background distributions where the drug AUCs were randomly shuffled ($N$ = 8,192) among the cell lines for each drug before running the PLS algorithm (Fig 3C). Of 27 tested drugs, the PLS model predicted AUC significantly above background ($P$ < 0.05) for 12 drugs, of which 10 drugs had adjusted $P$ < 0.05. For four drugs, the Spearman correlation exceeded $\rho_s$ > 0.5, namely, the MAPK pathway inhibitor dabrafenib, the cytotoxic drugs MMAE and DM1, and the HDAC inhibitor trichostatin A. For some drugs, a negative Spearman rank

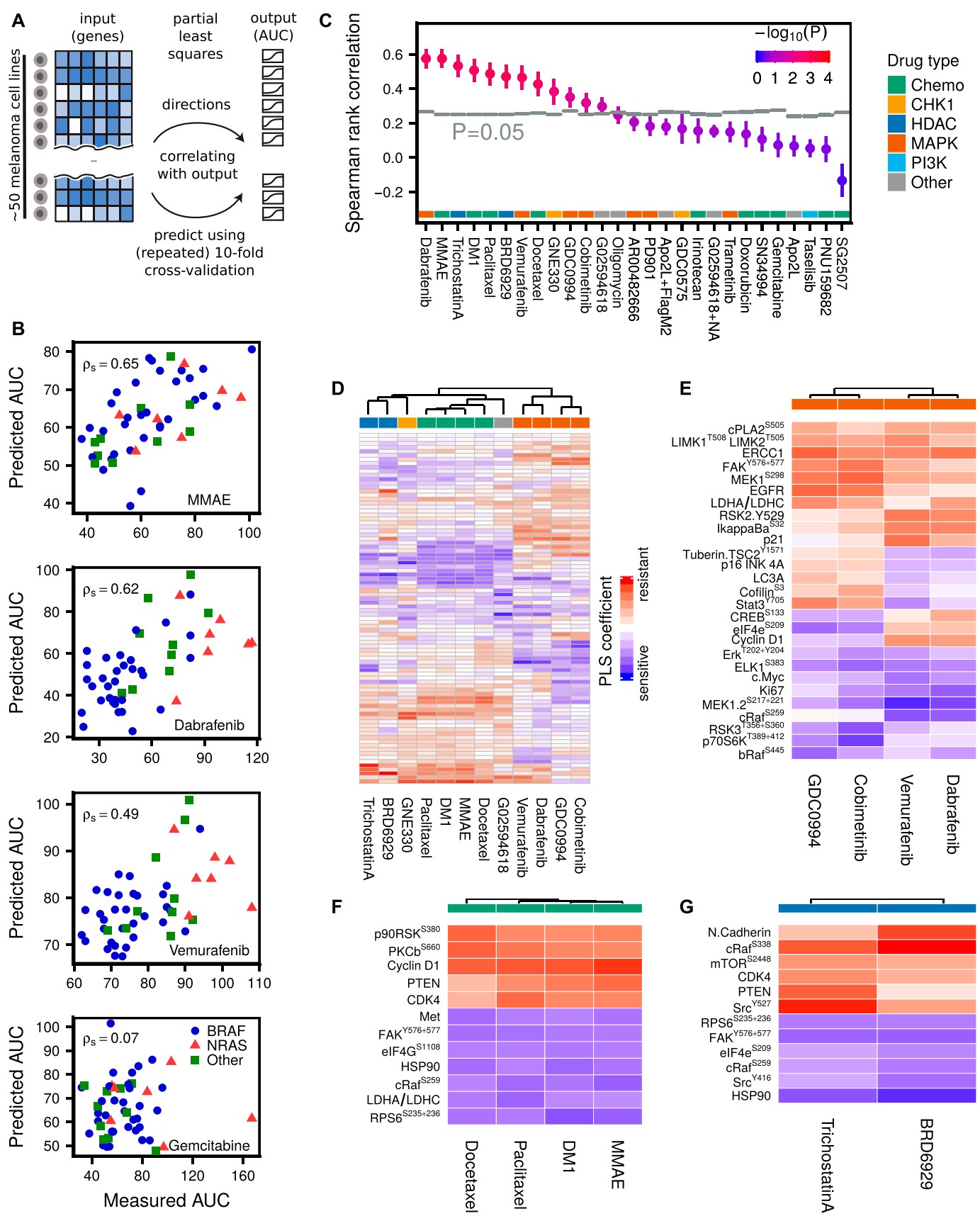

correlation was observed, indicating a nonpredictive model after cross-validation (see Supplemental Information, Section 1).

PLS produced for each drug a linear model from proteomic state to AUC. For normalized proteomic data, the magnitude of each linear coefficient can be taken as a proxy of the relative importance of the corresponding (phospho)protein (see Figs 3D and S3–S5 and Supplemental Table T1). Interestingly, the PLS coefficients clustered according to drug type. For MAPK pathway inhibitors, MEK inhibitors and BRAF inhibitors formed distinct but similar clusters. Fig 3E depicts the most important markers for these inhibitors, as well as the most differential markers between MEK and BRAF inhibitors. Increased levels of Elk-1$^{S383}$, MEK1/2$^{S217+221}$, Erk1/2$^{T202+Y204}$, and cMyc were associated with drug sensitivity, whereas increased levels of EGFR, ERCC1, MEK1$^{S298}$, and FAK$^{Y576+577}$ were associated with drug resistance. The observed correlation of Elk-1$^{S383}$, a direct target of ERK, and MEK double phosphorylation with sensitivity to MAPK inhibitors merely reflects the expected effect of increased activity when MAPK signalling is more active. Among the resistance markers to MAPK inhibitors, high expression of EGFR, which is normally expressed in very low levels in melanoma, is a known resistance mechanism to counteract BRAF and MEK inhibitors (Klinger & Blüthgen, 2014; Sun et al, 2014). MEK1$^{S298}$ is reduced by active ERK through phosphorylation of the neighbouring site T292 on MEK (Eblen et al, 2004), thus this mark indirectly indicates low ERK activity and consequently resistance to drugs inhibiting the RAF/MEK/ERK cascade. FAK has been directly linked to MEK inhibitor sensitivity in melanoma cell lines (Rožanc et al, 2018). Stat3$^{Y705}$ and Cofilin$^{S6}$ were among the group of markers that were associated with sensitivity to BRAF inhibitors, but resistance to MEK inhibitors. In contrast, higher expression of CREB$^{S133}$ and eIF4e$^{S209}$ were associated with sensitivity for MEK inhibitors, but resistance to BRAF inhibitors. Stat3$^{Y705}$ activation has previously been linked with resistance to MEK inhibition in lung cancer cell lines (Dai et al, 2011). For cytotoxic or HDAC drugs, increased levels of FAK$^{Y576+577}$, HSP90, and RPS6$^{S235,236}$ were associated with drug sensitivity, whereas increased levels of PTEN and CDK4 were associated with drug resistance (Fig 3F and G).

Apart from PLS, we tried five other algorithms to predict drug sensitivity: PLS2 (multi-output), elastic net, Lasso, regression tree, and a "maximum correlation" model, which performs linear regression on the variable that is most correlated with drug sensitivity. We found PLS to be the best performing method among the six for various datasets and, therefore, decided to use it as the primary method (see the Materials and Methods section and Figs S6 and S7).

PLS predicts drug sensitivity from a linear combination of input data that more accurately describes the state of the cell than individual signals, which also leads to a reduction of noise by averaging over multiple measurements. However, from a clinical perspective, a complex biomarker might be of limited use in practice. Hence, finding a balance between predictability and simplicity is an important issue to address. From cross-validated Lasso regression, it is possible to determine how the prediction quality is affected when limiting the number of nonzero coefficients. In Figs S8–S11, we show that for the drugs vemurafenib and MMAE one can use as few as 5–10 (phospho)proteins and still be close to optimal performance.

## Transcriptomic state predicts drug sensitivity for 11 of 27 drugs

The set of measured (phospho)proteins corresponds only to a small fraction of all genes expressed in the cell; therefore, we also decided to make drug sensitivity predictions from full transcriptomic states for comparison. We acquired transcriptomic (RNAseq) data for 46 of 49 cell lines in our panel, covering 26,378 genes (Supplemental Data D4). Running PLS on mean+variance filtered transcriptomic data (6,286 genes) yielded 11 drug AUC predictions above randomized background ($P < 0.05$; $N = 1,024$), eight of which had adjusted $P < 0.05$. The most predictive ($\rho_s > 0.05$) drugs were the HDAC inhibitor trichostatin A, and cytotoxic drugs MMAE and paclitaxel (Fig 4A). The linear PLS models from gene expression to drug AUC can be found in Supplemental Table T2.

Of the measured genes, we expected a majority to be unrelated to drug sensitivity, merely feeding noise into the predictions. To reduce the input space, we tried various subselections, including using only genes reported in the COSMIC Cancer Gene Census (Forbes et al, 2015), genes present in the proteomic dataset, signature genes in key cellular pathways as reported in SPEED (Parikh et al, 2010) (H2O2, IL-1, JAK-STAT, MAPK, MAPK+PI3K, PI3K, TGFb, TLR, TNFa, VEGF, and Wnt), using mean+variance filtering, and using randomly selected genes as a control. Somewhat surprisingly, all subselections, except random selection of fewer than 500 genes, showed on average similar predictiveness of drug sensitivity (Figs S12 and S22). For the remainder of this report, we use mean + variance filtered transcriptomic data for drug sensitivity predictions. In the case of transcriptomic data, we found that PLS and PLS2 outperformed elastic net and Lasso for most drugs (Fig S7 and see the Materials and Methods section).

Our results show that transcriptomic and proteomic states are about equally predictive of drug sensitivity in our melanoma cell line panel, both being more predictive than genomic state, which was only predictive of BRAF inhibitor response. The proteomic data were more predictive of MAPK pathway inhibitor sensitivity than the transcriptomic data. Combining the transcriptomic and proteomic

---

**Figure 3. Predicting drug sensitivity from proteomic data using PLS.**
**(A)** PLS predicts drug AUC from proteomic (RPPA) data by defining new variables as linear combinations of the original (phospho)proteins, such that the new dimensions are maximally correlated with AUC. The agreement between measured and predicted AUC after repeated 10-fold cross-validation was quantified using Spearman rank correlation. **(B)** Selected drug AUC predictions for MMAE (cytotoxic), dabrafenib (BRAFi), vemurafenib (BRAFi), and gemcitabine (cytotoxic, failed prediction). Predictions are averaged over 100 repeats. **(C)** Drug AUC predictions compared with randomly shuffled background distributions ($N = 8,192$). **(D)** PLS defines a linear model from proteomic data to AUC for each drug. The (normalized) PLS coefficients suggest associations between (phospho)proteins and drug sensitivity or resistance. Unsupervised hierarchical clustering of the linear drug sensitivity models predominately clusters drugs sharing common targets. **(E)** Most important markers for MAPK inhibitors, as well as the most important differential markers between MEK and BRAF inhibitors. **(F)** Most important markers for cytotoxic drugs. **(G)** Most important markers for HDAC inhibitors.

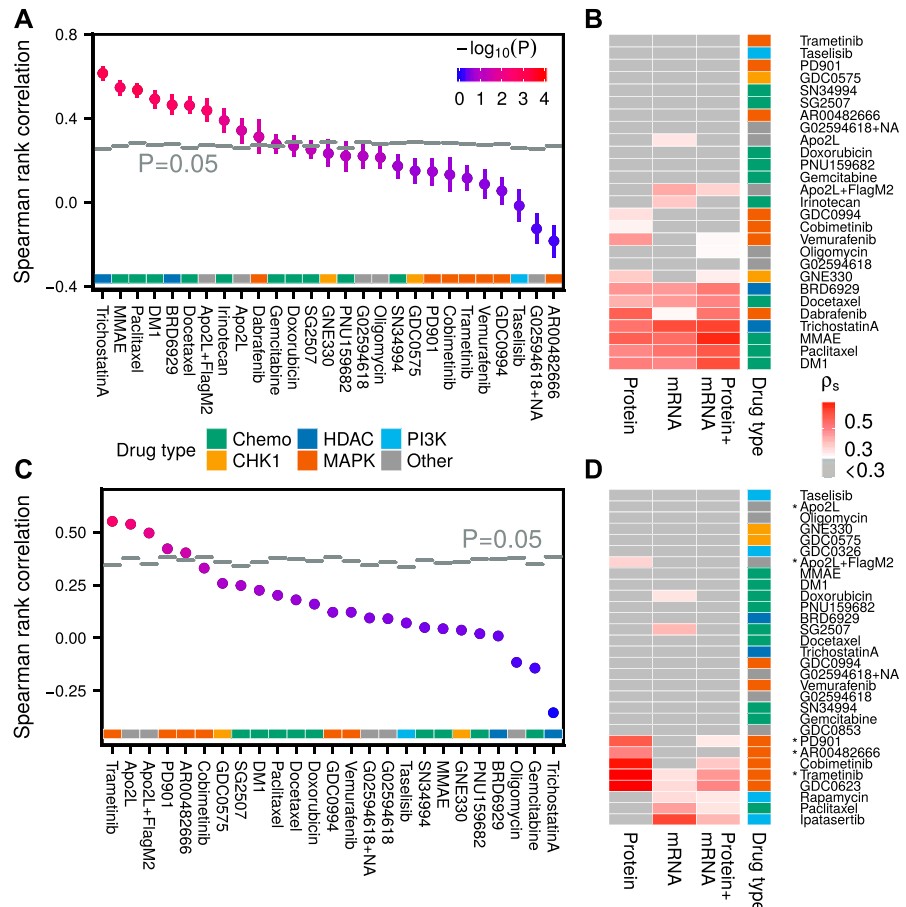

**Figure 4. Comparing drug sensitivity predictions from different data types in melanoma and endometrial cancer cell lines.**

**(A)** Drug sensitivity predictions from transcriptomic data compared with randomly shuffled background distributions (N = 1,024). **(B)** Drug sensitivity predictions based on transcriptomic, proteomic, or combined (late integration) data in melanoma cell lines. Only cell lines, which had data of all three types genomic/transcriptomic/proteomic were used for model building. **(C)** Drug sensitivity predictions in endometrial cancer cell lines based on models trained in melanoma, using only proteomic signals measured in both datasets. **(D)** Drug sensitivity predictions based on proteomic, transcriptomic, or combined data in endometrial cancer cell lines. Drugs which were predictive in the melanoma to endometrial cross-model in (C) are marked with an asterisk.

models, by either simply augmenting the input before running PLS (early integration) or by weighting the two resulting PLS models by their inverse RMS (late integration), resulted in drug sensitivity predictions, which were typically between the two separate models (Figs 4B, S13, and S14). Because there was a noticeable co-occurrence of drugs that could be predicted from either transcriptomic or proteomic data, we reasoned that the two datasets likely were correlated. Indeed, in Fig S15, we show that many (phospho)proteins could be predicted from transcriptomic data using PLS, but only a handful using genomic data (Fig S16).

### Proteomic and transcriptomic state predict drug sensitivity for selected drugs also in endometrial cancer cell lines

We acquired a complementary dataset of 27 endometrial cancer cell lines, with transcriptomic, proteomic, and pharmacological (22 matched cell lines) data (Supplemental Data D1, D2, and D4). From 109 drugs, we selected the 30 drugs with the highest coefficient of variation in AUC across the cell line panel. Proteomic PLS models yielded accurate drug AUC predictions especially for MEK inhibitors (Fig 4D). Transcriptomic PLS models were on average less predictive but also covered additional drug types.

From 60 measured (phospho)proteins and 30 drug sensitivities in the endometrial cancer panel, 39 (phospho)proteins and 25 drug

sensitivities were also measured in the melanoma panel. To test whether drug AUC models could be transferred between these two tumor types, we trained models in melanoma, using the shared (phospho)proteins, and tested them in endometrial data. As a control, we randomly shuffled endometrial AUCs (N = 1,024) before applying the transferred models. We found five drugs (trametinib, Apo2L, Apo2L+FlagM2, PD901, and AR00482666), which could be predicted by the cross-melanoma models (Fig 4C). Four of these drugs belonged to the six most predictive drugs when using within-tumor–type endometrial proteomic models (Fig 4D), suggesting that to a certain degree, cross-cancer predictability is possible. These results should, however, be interpreted with caution as the two best cross-cancer drug sensitivity predictions had only borderline significant corrected P-values (P ≈ 0.05), and the transferred drugs were not predictive in melanoma to start with.

### Proteomic and transcriptomic states are more predictive of drug sensitivity than genomic state in eight additional tumor types using Cancer Cell Line Encyclopedia (CCLE) data

To assess the generality of our findings beyond melanoma and endometrial cancer cell lines, we took genomic, transcriptomic, and proteomic data from the CCLE (Barretina et al, 2012) and matched with drug sensitivity data from the Cancer Therapeutics Response

Portal (Seashore-Ludlow et al, 2015) (CTRP, 481 drugs and 561 matched cell lines) and the CCLE (Barretina et al, 2012) (24 drugs and 424 matched cell lines). Drugs with low coefficient of variation ($\sigma/\mu < 0.2$) across cell lines were filtered out.

We built PLS (transcriptomic/proteomic data) and regression tree (genomic data) models separately for the eight tumor types with the largest number of cell lines, as well as pan-cancer models (Figs 5A and B, and S21). Only cell lines that had corresponding data of all three types (genomic/transcriptomic/proteomic) were used for model building. The volcano plot in Fig 5A shows that transcriptomic and proteomic data predict drug sensitivity significantly above randomly shuffled background ($N$ = 1,024) for many CCLE and CTRP drugs, whereas genomic data were much less predictive. Differences between transcriptomic and proteomic data were comparatively small, with only a slight edge for the transcriptomic data (see also Fig S17C). The separation between genomic and transcriptomic/proteomic predictions in Fig 5B and C is exaggerated at the low end by the fact that the two learning approaches, regression trees and PLS, have different background distributions for randomly shuffled data, in particular when measured by Spearman rank correlation (see Supplemental Information, Section 1). We, therefore, also show the corresponding plots of Fig 5B when using the fraction of variance explained measure instead, as well as $P$-value distributions (Figs S18–S20). Again, we find that genomic data are less predictive of drug sensitivity than transcriptomic/proteomic data.

### Transcriptomic and proteomic—but not genomic—data are more predictive of drug sensitivity than tumor type in pan-cancer analyses

To determine whether genomic/transcriptomic/proteomic data in CCLE provided additional information about drug sensitivity not already encoded in the tumor type, we compared the genomic/transcriptomic/proteomic models with using tumor type alone as a predictor (Fig 5C). To predict drug sensitivity from tumor type, we left one cell line out at the time, computed the mean sensitivity of the remaining cell lines for the tumor type, then assigned this value to the left out cell line. Transcriptomic/proteomic PLS models were on average more predictive than tumor type alone, indicating that proteomic and transcriptomic data encode additional useful information for drug sensitivity predictions, whereas genomic data on average were less predictive than tumor type. Interestingly, the top predictions from genomic data across tumor types were all MEK inhibitors (PD0325901, selumetinib, and trametinib), which were associated with BRAF and KRAS/NRAS mutations.

### Drug sensitivity models can be transferred between tumor types, but after correcting for differences in cell line numbers that the models were trained on, the transferred models perform worse than within-tumor–type predictions

To see if drug sensitivity models could be transferred between tumor types, we excluded all cell lines belonging to a given tumor type and trained drug sensitivity models using the remaining cell lines in CCLE. The resulting models were subsequently applied to the excluded cell lines and the Spearman rank correlation between predicted and measured drug sensitivity computed. Our results show that cross-models were predictive for many drugs using either transcriptomic or proteomic data (Fig 5D, Cross), with similar performance compared with within-tumor–type predictions (Fig 5D, Within). Notable exceptions were pancreatic (and to lesser degree ovarian) cancer cell lines, where cross-model predictions surpassed within-tumor–type predictions, and haematopoietic and lymphatic cancer cell lines, where the cross-models consistently performed poorly. This might reflect the fact that cancers of the haematopoietic and lymphatic system are molecularly different from tumors of epithelial or neuronal origin. Because it is possible that the performance of the cross-tumor–type predictions could be attributed solely to the much larger training set of cell lines, we also built cross-models that were trained on the same number of cell lines as the target tissue, by randomly selecting cell lines from other tissue types ($N$ = 1,024 repeats). After correcting for cell line number differences, the cross-tumor–type models performed, as expected, worse than the within-tumor–type predictions (Fig 5D, Cross fair).

To reduce the computational time, the transcriptomic data were reduced to genes reported in the COSMIC Cancer Gene Census (Forbes et al, 2015). According to Figs S12 and S22, this gene subset predicts drug sensitivity equally well as compared with unfiltered transcriptomic data both in our melanoma panel and in CCLE data.

## Discussion

Biomarkers are key determinants of success in precision medicine; however, many currently used targeted therapies still lack predictive biomarkers. Pharmacogenomic studies have been instrumental to derive biomarkers (Barretina et al, 2012; Garnett et al, 2012; Cook et al, 2014; Costello et al, 2014; Klijn et al, 2014; Haverty et al, 2016; Iorio et al, 2016); yet, genomic biomarkers are largely limited to mutations that are directly targeted by the drug, and only a handful have been clinically validated, raising the question whether other biological data could be more predictive of drug sensitivity (Costello et al, 2014; Yuan et al, 2014). In this work, we systematically compared (basal) genomic, transcriptomic, and proteomic data as predictor of drug sensitivity, and found transcriptomic and proteomic data to be the best predictor for most drugs, both in an in-house melanoma cell line panel and in eight additional tumor types of the CCLE. Tumor cells acquire their malignant phenotype through a series of (epi)genomic alterations that activate proliferative signaling pathways (Hanahan & Weinberg, 2011). In this way, tumor cells can reach their final malignant state through a plethora of genomic paths in a micro-evolutionary process, and this redundancy might obscure underlying associations with drug sensitivity. Our results suggest that transcriptomic and proteomic data are projections of genomic data, which are more immediately linked to the biological state, thus making them stronger biomarkers for drugs that do not directly target activating mutations, in agreement with a recent breast cancer cell line panel study (Costello et al, 2014). Clinically, transcriptomic classifiers are used in breast cancer treatment to stratify subtypes and identify risk groups (Parker et al, 2009). Also in

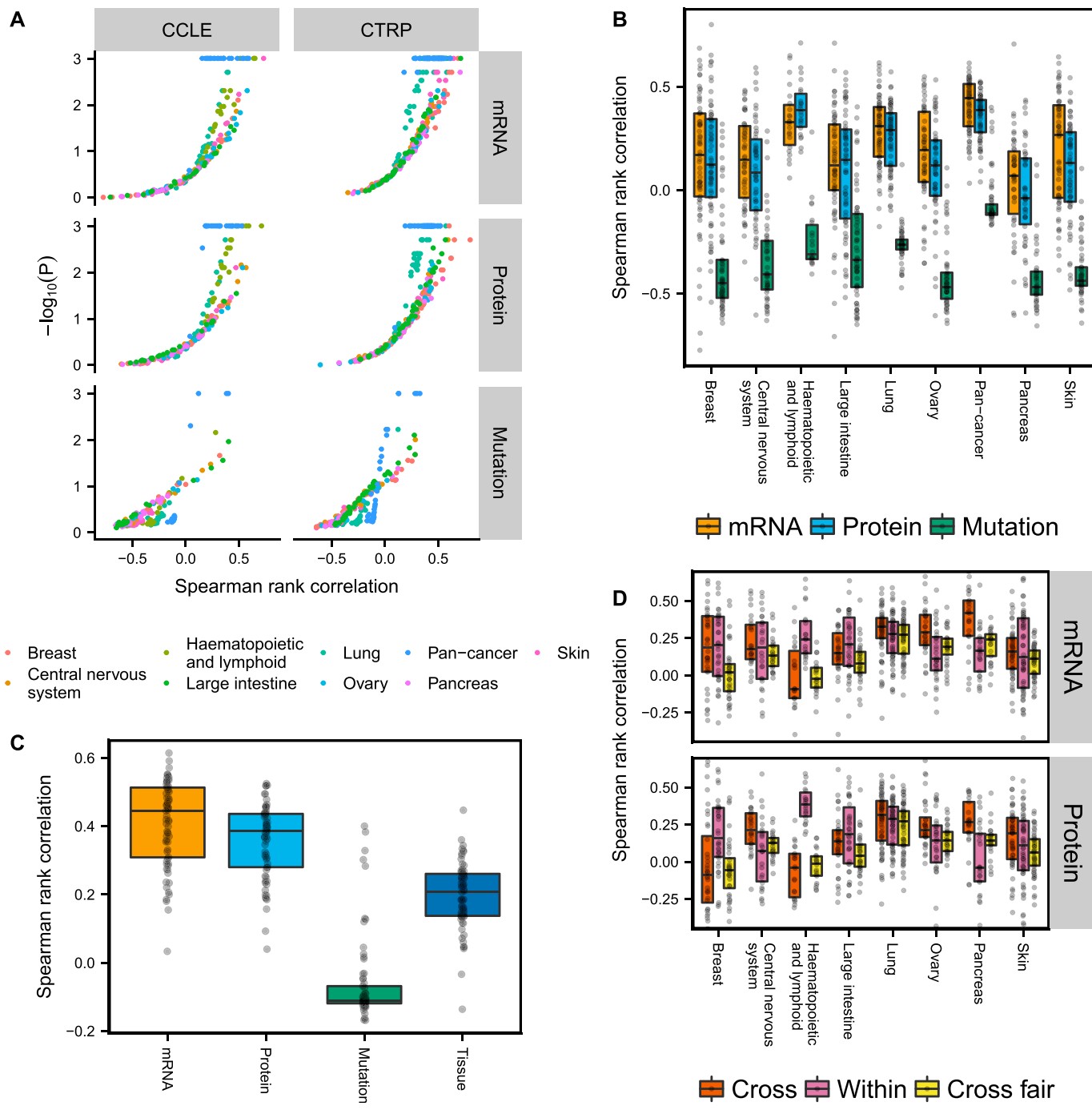

**Figure 5. Drug sensitivity predictions from CCLE data.**
**(A, B)** Spearman rank correlation after cross-validation between predicted and measured drug sensitivity. The two different drug sensitivity censuses are combined in (B–D) by averaging the Spearman rank correlations for drug sensitivities predicted in both censuses. **(C)** Pan-cancer drug sensitivity predictions using either genomic/transcriptomic/proteomic data or tumor type. **(D)** All cell lines belonging to the tumor type specified by the x-axis were excluded and the remaining cell lines used for building drug sensitivity models (Cross). The models were then tested in the excluded cell lines. Alternatively, the cross-tumor–type models were trained on an equal number of randomly selected cell lines as the target tissue (Within) to make the comparison fair (Cross fair). Drugs with low coefficient of variation ($\sigma/\mu < 0.2$) either across the target tissue cell lines or the remaining cell lines were filtered out.

melanoma, recent work has shown that transcriptomic-based signatures can successfully predict outcome of single drug versus combinatorial treatment in four clinical trials (Wongchenko et al, 2017).

A main challenge of identifying potential biomarkers from cell line panels is the mere problem of dimensionality: there are many more possible biomarkers than cell lines. Pooling cell lines from multiple tissues together has been one strategy to gain statistical

power; however, this strategy requires that biomarkers are transferable between tumor types. When attempting to predict drug sensitivity in endometrial data from models that were trained in melanoma, we found that some drug sensitivity models could be transferred; however, with worse performance compared with within-tumor–type predictions. Analogously, we found that for most tumor types in CCLE, there was no or only marginal benefit to train models across other tumor types instead of training within the given tumor type, despite the much larger number of cell lines available for training.

Biomarkers need to be evaluated separately for each individual tumor type, as they are clinically required to stratify patients with a specific tumor type into subgroups. Our study, as well as previous studies (Iorio et al, 2016), show that there are only few genomic biomarkers derived from pan-cancer models that are predictive, and most markers are not more predictive than tumor type alone, thus limiting their clinical relevance. In contrast, transcriptomic and proteomic data may provide useful additional information that go beyond the tumor type for drug sensitivity predictions.

Many important biomarkers have been discovered in cell line panels. However, cell line panel studies have their limitations as they fail to capture many essential aspects of tumor development, for example, processes related to intercellular communication and immune infiltration/activation, and stromal interactions. Therefore, follow-up studies on tumor samples are necessary to determine how well transcriptomic and proteomic biomarkers perform clinically, and whether cell line–derived models are directly transferable. Using organoid-based "living biobanks" to validate complex transcriptomic and proteomic biomarkers will be a next logical step to move model systems that are amenable to screening experiments closer to the clinical reality (van de Wetering et al, 2015; Schütte et al, 2017).

Given that ~40–50% of BRAF$^{V600E/K}$ melanoma patients fail to respond to BRAF inhibitors (McArthur et al, 2014), there is room for improvement in the biomarker-driven selection of patients beyond simple genetic testing. Recent work shows that transcriptomic biomarkers can identify subgroups responding to combinatorial BRAF+MEK inhibitor treatment (Wongchenko et al, 2017), but for those 50–60% of melanoma patients without BRAF$^{V600E/K}$ mutation, other suitable therapies are needed. Using transcriptomic or proteomic states as predictor of drug sensitivity could, if clinically validated, help improve patient/drug matching and guide the development of new therapeutics suitable for other groups of cancer patients.

# Materials and Methods

### Next-generation sequencing data acquisition and analysis

Cell line screening and next-generation sequencing analysis were performed as described previously (Haverty et al, 2016). Briefly, cell lines were obtained from a variety of academic sources (i.e., ATCC and DSMZ) and drugs were obtained from in-house synthesis or purchased from commercial vendors.

Gene expression levels were quantified by RNASeq via Illumina sequencing (75-bp paired-end reads) in one sample per cell line.

Reads were aligned to the genome (GRCh37.1) using GSNAP (Wu & Nacu, 2010). Reads overlapping gene exonic regions were counted and normalized to gene size and library size as reads per kilobase per million.

DNA mutations were assessed by Illumina exome sequencing (75-bp paired-end reads) in one sample per cell line. Only mutations from genomic locations covered by at least 30 reads, where at least 20% of the reads supported the given mutation, were considered. Mutations were filtered for known population variants in the 1,000 Genomes Project (Auton et al, 2015) (African, American, Asian, East Asian, European, South East Asian, and combined populations) and NHLBI Exome Sequencing Project (http://evs.gs.washington.edu/EVS/) (African American and European American populations), using VEP filtering tools with a frequency threshold of 1%. Mutations where additionally filtered for severity by ignoring mutations with an impact score labeled as less than MODERATE according to the VCF format specification (Danecek et al, 2011). Mutations in the known "false positive cancer genes" HLA- and MUC- were discarded (Lawrence et al, 2013).

### Reverse-phase protein array (RPPA) acquisition and analysis

Cells were lysed in a buffer containing tissue protein extraction reagent (T-PER; Thermo Fisher Scientific), 300 mm NaCl, and protease and phosphatase inhibitors (Sigma-Aldrich). The samples were assessed by RPPA analysis (Theranostics Health) using 88 validated antibodies. Replicate samples were printed onto nitro-cellulose slides in four separate quadrants. Total protein was measured by SYPRO Stain, and the intensities of specific antibody signals were subtracted from secondary antibody signal and normalized to the total protein (to account for differences in protein content between samples). The data from each slide were normalized to the median of each quadrant to compensate for spatial effects.

### Drug sensitivity screening

Cell viability assays were performed as described previously (Haverty et al, 2016). Nine drug doses following an evenly (log) spaced 1:3 serial dilution were used. The minimum/maximum concentrations were adjusted for each drug depending on potency. Cell viability was measured by CellTiter-Glo (Promega) following 72 h of drug treatment. Throughout the article, the abbreviation NA denotes nicotinic acid.

### Predicting drug sensitivity from transcriptomic or proteomic data using PLS

PLS is a statistical method designed for performing regression analysis when the number of predictors is larger than the number of observations, a regime where traditional linear regression breaks down. This is a situation commonly encountered in modern biological experiments. We used the R package pls to predict drug sensitivity from normalized proteomic (RPPA) or transcriptomic (RNAseq) data, and validated the model using repeated 10-fold cross-validation. The optimal number of PLS components was determined by nested 10-fold cross-validation. See algorithm

outline below (Source code in Supplemental Information). For each drug:

0. Discard cell lines without sensitivity data.
   1. Leave 10% of the cell lines out for testing (*test data*). Denote remaining cell lines *training data*.
   2′. RNAseq data: Transform expression values as $\log_2(1 + x)$. Discard genes with a mean expression below 2.0 or SD below 0.5 across the cell lines.
   2′. RPPA data: Transform expression values as $\log_2(x)$. Discard genes with SD below 0.1 across the cell lines.
   3. Normalize and center training data.
   4. Run R pls::plsr function on training data using 1–10 PLS components (PC:s).
   5. Choose the optimal number of PC:s as the lowest number PC where increasing PC by one, PC → PC+1, yields a higher RMS between predicted and measured drug sensitivity ($PC_{min}$ = 1, $PC_{max}$ = 10), as determined from nested 10-fold cross-validation.
   6. Scale and shift *test data* using the same normalization and centering parameters as for the *training data* in step 3. Predict drug sensitivity of *test data* using the optimal model from step 5.
   7. Iterate step 2–6 for all 10-folds.
   8. Compute Spearman rank correlation between predicted and measured drug sensitivities.
   9. Repeat 10-fold cross-validation multiple times. Average Spearman rank correlations between predicted and measured drug sensitivity.

In addition to mean + variance expression filtering, we also tried to reduce transcriptomic data using only genes reported in the COSMIC Cancer Gene Consensus, genes present in the RPPA dataset, random genes, or signature genes in key cellular pathways as reported by SPEED (Parikh et al, 2010), namely, H202 (60 signature genes), IL-1 (141), JAK-STAT (114), MAPK (559), MAPK+PI3K (118), PI3K (67), TGFb (142), TLR (181), TNFa (259), VEGF (56), and Wnt (83). The default SPEED settings were used to determine whether a gene was considered differentially expressed or not (first percentile z-score, 50[th] percentile expression level, 20% overlap across experiments).

Instead of using regression coefficients as gauge of variable importance, we also tried the alternative measure Variable Importance in Projection (Mehmood et al, 2012) (VIP), often used in PLS. VIP, however, did not show an advantage over regression coefficients, which we decided to use for the benefit of interpretability.

To parallelize drug sensitivity predictions, we used the workflow tool Snakemake (Köster & Rahmann, 2012). Many analyses were computationally expensive, in particular, the drug sensitivity predictions from transcriptomic data with randomly shuffled background distributions in Fig 5, which took on the order of $10^5$ CPU hours to perform.

### Predicting drug sensitivity using other machine learning algorithms

In addition to PLS, five additional learning algorithms were used for comparison. The general algorithm outlined for PLS above was followed for all learning methods with minor adjustments (Source code in Supplemental Information). *PLS with multiple outputs:* An alternative version, PLS2, allows all outputs to be predicted at once, which in the case of correlated outputs might lead to improved predictions. On the other hand, the number of PLS components needs to be "compromised" between all drugs, which can impact the predictions negatively. We chose the optimal number of PLS components by minimising the summed RMS between predicted and measured drug sensitivity for all drugs. *Elastic net:* This is an extension of ordinary linear regression where the sum of squared errors (the cost function) is augmented by a penalty term $\lambda[(1 - \alpha)\|\beta\|^2 + \alpha\|\beta\|_1]$ to reduce the risk of overfitting. The penalty term is weighted by a scale factor $\lambda$, which is typically determined from cross-validation. We performed elastic net regression using the R package glmnet (Friedman et al, 2010) to predict drug sensitivity. The model was validated using (repeated) 10-fold cross-validation, and the penalty weight determined from nested 10-fold cross-validation (Source code in Supplemental Information). Eleven different values of $\alpha$ = 0.0, 0.1, …, 1.0 were tested in the nested cross-validation step. By successively increasing the penalty $\lambda$, one can identify key variables for predicting drug sensitivity. *Lasso regression:* Elastic net regression with $\alpha$ = 1. *Regression tree:* We used the R package rpart (Therneau & Atkinson, 2018) to build drug sensitivity regression trees and used 10-fold nested cross-validation with a minimum bucket size of five to find the optimal tree pruning. *"Maximum correlation" model:* We performed linear regression on the single variable with largest (absolute) correlation with drug sensitivity and validated the model using (repeated) 10-fold cross-validation.

### Hierarchical clustering

Unsupervised hierarchical clustering was performed using complete-linkage clustering over a Euclidean metric.

## Data Availability

All in-house data and source code are available in Supplemental Information. Data for Fig 5 is publicly available (Barretina et al, 2012; Seashore-Ludlow et al, 2015; Li et al, 2017). Raw sequencing data are available at the European genome-phenome archive under the accession number EGAS00001000610.

## Supplementary Information

## Acknowledgements

The authors wish to thank Marie-Claire Wagle at Genentech for assistance with the experimental procedures and Clemens Messerschmidt at the Berlin Institute of Health Core Unit Bioinformatics for assistance with

preprocessing exome sequencing data. N Blüthgen acknowledges funding from BMBF/PTJ project ColoSys, Berlin Institute of Health, and Roche/Genentech.

## Author Contributions

M Rydenfelt: conceptualization, data curation, software, formal analysis, investigation, visualization, methodology, and writing—original draft.
M Wongchenko: conceptualization, resources, investigation, and writing—review and editing.
B Klinger: conceptualization, investigation, and writing—review and editing.
Y Yan: conceptualization, resources, supervision, funding acquisition, and project administration.
N Blüthgen: conceptualization, supervision, funding acquisition, investigation, visualization, project administration, and writing—review and editing.

## Conflict of Interest Statement

The study was financed in part by a collaborative research grant from Roche/Genentech. M Wongchenko and Y Yan are employees of Genentech, Inc., a member of the Roche group. M Wongchenko and Y Yan have equity interest in Roche.

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
