## [Reviewer comments · Life Science Alliance]

Life Science Alliance

The cancer cell proteome and transcriptome predicts sensitivity to targeted and cytotoxic drugs

Mattias Rydenfelt, Matthew Wongchenko, Bertram Klinger, Yibing Yan, and Nils Blüthgen
DOI: <https://doi.org/10.26508/lsa.201900445>

Corresponding author(s): Nils Blüthgen, Charite Universitätsmedizin Berlin; Nils Blüthgen, Charite Universitätsmedizin Berlin; and Yibing Yan, Genentech Inc.

Review Timeline:

Submission Date:	2019-06-03
Editorial Decision:	2019-06-04
Revision Received:	2019-06-13
Accepted:	2019-06-14

Scientific Editor: Andrea Leibfried

Transaction Report:

Please note that the manuscript was previously reviewed at another journal and the reports were taken into account in inviting a revision for publication at *Life Science Alliance* prior to submission to *Life Science Alliance*.

We thank both reviewers for their constructive and valuable comments, which helped to improve our manuscript. We have carefully and extensively revised the manuscript and performed a large number of additional analyses. The most important changes/additions are:

- (1) Compare genomic/transcriptomic/proteomic data as predictor of drug sensitivity in eight additional tumor types using data from the Cancer Cell Line Encyclopaedia (CCLE). The new CCLE analysis substantiates that proteomic and transcriptomic data can be used to predict drug sensitivity for a variety of drugs, whereas genomic data predict drug sensitivity only for a narrow selection of drugs, in agreement with our in-house melanoma panel
- (2) Investigate "transferability" of drug sensitivity models between tissue types also in CCLE data
- (3) Compare different machine learning algorithms, namely elastic net, Lasso, PLS, PLS with multiple outputs, regression trees, and a maximum correlation method
- (4) Compare different strategies to reduce high-dimensional transcriptomic data both in our in-house melanoma panel and CCLE data
- (5) Interpret our results strictly conservatively, which is also reflected in the updated title of the manuscript.

Please find a point-by-point response below.

Reviewer #1:

This manuscript is interesting from the point of view of trying to infer drug sensitivity/resistance from the activation status of signaling pathways operating in cancer cells.

However, there are major issues with the work that preclude publication in its current form and a lot of work would be required in order to make it fit for publication.

The main issues are:

The authors are drawing a lot of bold and broad conclusions from very limited data. As an example, the authors state: "Our results suggest that evaluation of cell signaling prior to drug treatment may be sufficient to predict tumor response". A second example: "...showing that signaling data can predict drug viability." The authors should not present their interpretation of the data as if they were biological facts because, unfortunately, the authors do not follow up their hypothesis with experimentation that would underscore the above notions. As much as this reviewer likes the concept, the conclusion is not backed by the data.

We agree that certain statements in our original submission were not carefully phrased. To address the reviewer's comment we now compare genomic/transcriptomic/proteomic data as predictor of drug sensitivity in eight additional tumor types using publicly available data in CCLE, to test if our findings in melanoma can be extended to other tumor types (see Figure 5). In addition, we now

interpret our results strictly conservatively, which is also reflected in the updated title of the manuscript.

The number of RNAs, proteins and p-proteins that were used in this study are very limited. This reviewer doubts that broad conclusions can be drawn from such a limited set. It is further very unclear why the authors only used 200 mRNA species (starting from 26,000) and the criteria for filtering appear arbitrary.

We fully agree that feature selection is a crucial part of machine learning. In the revised manuscript, we now systematically assess various strategies for reducing high-dimensional transcriptomic data before making drug sensitivity predictions, both using our in-house melanoma panel and CCLE data (see Supplemental Figures S12+S21). An interesting outcome is that the PLS drug sensitivity predictions are rather insensitive to the precise selection of genes, presumably due to correlations between groups of genes.

No comparison is provided as to how the author's method performs vs other established ones (such as elastic net analysis or some such).

See next comment

The authors should also better explain why they used PLS to model drug effects. The argument of " $p > n$ " is not a valid one. If they had used the entire drug sensitivity matrix as input, PLS would have been the natural choice. But because they used each individual drug as input, (penalized) linear regression, like LASSO or elastic net, would have been better choices. Overall, the chosen method complicates interpretability and thus it remains unclear how good the models really are and which may have been a reason why the authors did not attempt follow-up experiments to prove their point.

There are two classes of PLS: PLS1 and PLS2, that find relations between X ($n \times p$ dimensional) and Y , where Y is a vector (PLS1) or a matrix Y (PLS2). PLS1 almost always results in more predictive models as the X decomposition can be optimised for just one output variable – without compromising with the other output variables (Reference "Multivariate Data Analysis" by Kim H. Esbensen, page 150). PLS regression is particularly designed for situations where the matrix of predictors has more variables, some of which might be correlated, than observations (Reference: "PLS-regression: a basic tool of chemometrics", Wold et al. 2001). Both of these conditions are unquestionably true in our analysis, where $n=49$ cell lines in the melanoma panel, and $p > 49$, with strong correlations between different X columns. Thus, PLS is one natural method to try, among others. Notice that the " $p > n$ " condition holds irrespectively of whether we choose PLS1 or PLS2.

To address the reviewer's comment, we now include a thorough comparison of different machine learning algorithms, namely elastic net, Lasso, PLS, PLS with multiple outputs, regression trees, and a maximum correlation method, which can be found in Supplemental Figure S6+S7 for proteomic and transcriptomic data

respectively. We find that PLS performs similarly well as elastic net and Lasso in proteomic data, but better than these algorithms in transcriptomic data, and hence we decided to use PLS as our primary method. Elastic net, which is a generalisation of Lasso, interestingly reduced to Lasso for our dataset.

Reviewer #2:

The authors generated (phospho)proteomics and mutation data as well as drug response data for 109 drugs on a panel of 43 melanoma cell lines. Multiple analyses were performed and many broad claims are made. These include 1) that 'proteomic state is more predictive of drug sensitivity than transcriptomic state'; 2) that tissue type is equally predictive of response in the pan cancer setting as molecular data that 3) a proteomics predictor trained on one cancer type (melanoma) can not be transferred to another cancer type (endometrial) and that 4) on drug proteomics measurements are not more predictive for response than data collected in the treatment naive setting for a specific drug.

Major comments

1. While the study looks at potentially interesting problems fairly strong claims are made as outlined above and these can not be fully supported by the data presented. Here are some specifics:

1.1 The claim that proteomics is more predictive than mRNA can, so some degree be made based on the data in Panel 4b as there are more red cells (significant) in the proteomics column as in the mRNA column. However, there are only 27 drugs so adding a significance to this number will likely be hard. Second, each data type is predictive for a different set of drugs - admittedly there is also a large amount of overlap. This is a more accurate and more important message. It is quite unexpected that mRNA data is not predictive of Vemurafenib response. In addition, one might question the validity of such a comparison on melanoma which is so strongly driven by MAPK signaling. Finally, showing this in only melanoma is interesting, but the message would be far more relevant and strong if it is demonstrated in at least two additional cancer types (where the deck is not so obviously stacked in favor of phospho-proteomics).

To address the reviewer's comment we now compare genomic/transcriptomic/proteomic data as predictor of drug sensitivity in eight additional tumor types using publicly available CCLE data, showing that transcriptomic and proteomic data yield more accurate drug sensitivity predictions than genomic data for most drugs, similar to our in-house melanoma panel results (see Figure 5). In addition, we now interpret our results strictly conservatively, which is also reflected in the updated title of the manuscript.

We agree that there is a substantial overlap between transcriptomic and proteomic drug sensitivity predictions. We now correlate the proteomic and transcriptomic datasets, see answer to Comment 2 below.

1.2 The authors show, for a single drug (Dacomitinib), that the pre-treatment data is roughly equally predictive than the on-treatment data. Once again, this is an important hypothesis to investigate, but it is hardly convincing to show this for a single drug. This should be performed far more extensively to draw any meaningful conclusions. This is the type of result that gets cited out of context very easily. In this context, the LINCS project contains a far more exhaustive data set. In addition, while the performance may be similar, it would be interesting to know whether the post-treatment measurements show different phenomena than the pre-treatment data. For example, are the residuals correlated?

We agree that using one single drug is not enough to generalise the idea that post-treatment data is no more predictive than pre-treatment data. Additionally, when preparing this resubmission we became aware of that the Cobimetinib AUC predictions were sensitive to small perturbations in the input data, such as small changes in filtering settings or excluding individual cell lines from the analysis (see also answer to Comment 7). Since we are no longer confident in the results presented in this section, we have decided to completely remove it from the manuscript.

While the LINCS project has amazing data for a limited number of cell lines, our analysis requires data for a large number of cell lines of the same tumor type with matched drug sensitivity. We therefore believe that LINCS, in its current state, would not be the optimal route to compare post-treatment vs pre-treatment data. We would like to point out that upon submission, our proteomic dataset was, to the best of our knowledge, the largest available dataset for a cell line panel of a single tumor type with matched drug sensitivity data.

1.3 The comparison of the classifier on melanoma and endometrial cancer and the associated conclusion that classifiers can not be transferred from one cancer type to the next is of little value as it is a single case. Again, it is interesting to know, but based on these results one can hardly make a statement such as 'This confirms that signaling states are predictive of drug response, but that these, in general, are not conserved across tumor types'.

We agree that generalising this point requires additional data. To address the reviewer's comment we now investigate the transferability of drug sensitivity models for eight additional tumor types using both proteomic and transcriptomic data from CCLE (Figure 5e).

2. The authors present a new data type (phospho-proteomics, albeit limited in size). It will be very useful to relate this data type to the existing data types (mutations and RNAseq). For example, chart the associations between mutation and RNAseq data and phospho-proteomics. Which variables are correlated? What is the uniqueness and redundancy between the different data types? This is important to know as this provides a better stage for comparing the predictive capacity for drug response.

To address the reviewer's comment, we now correlate the proteomic and transcriptomic datasets. More precisely, we use PLS to predict proteomic markers from transcriptomic data, and find that the majority of proteomic markers can be rather well predicted from transcriptomic data (Supplemental Figure S15). This could explain why there is a substantial overlap in predictability of drug sensitivity from either the proteomic or transcriptomic dataset.

3. In the comparison of the mRNA and proteomics there are a number of differences that could explain the differences in performance. One of these is dimensionality. The authors do perform a selection of genes in specific pathways, but reducing the number of features in the mRNA expression dataset based on the variance in the features is a simple but necessary experiment. Since these data types will likely reveal the same biological phenomena employing different genes (MAPK activity is reflected by phosphorylation events in the pathway, while RNAseq will capture downstream expression programs) selecting the same genes/proteins is also an interesting approach but likely not the optimal one.

We fully agree that feature selection is a crucial part of machine learning. In the revised manuscript, we now systematically assess various strategies for reducing high-dimensional transcriptomic data before making drug sensitivity predictions, both using our in-house melanoma panel and CCLE data (see Supplemental Figures S12+S21). An interesting outcome is that the PLS drug sensitivity predictions are rather insensitive to the precise selection of genes, presumably due to correlations between groups of genes.

4. The authors do perform integration of RNAseq and proteomics data, but in the most simplistic way possible - early integration. It would be interesting to explore a number of interesting alternatives, including intermediate or late integration. In this way the complementarity of the data types will become more evident. Reliable biomarkers will likely emerge from both data types.

To address the reviewer's comment, we now perform also late integration, where the proteomic and transcriptomic models are weighted together by their respective inverse RMS error after cross-validation. Late and early integration resulted in similar performance (Supplemental Figure S13).

5. The authors use 'mean viability' as measure of drug response. If the drug concentrations are equally spaced, this is the same as the Area Under the dose-response Curve (AUC). If this is the case, the authors should rather refer to it as the AUC. If the concentrations are not equidistant, we suggest calculating the actual AUC and using this. Otherwise results should be repeated with the IC50.

The drug concentrations are equally spaced in log-space, hence we now follow the reviewer's advice and replace "mean viability" with "AUC" throughout the manuscript. We have also added a subsection in Methods describing the drug sensitivity screening experiments in greater detail (Line 372).

Minor comments:

6. Motivate why the Spearman correlation was used as an additional filter criterion in addition to Pearson correlation. Please also indicate how the R^2 is computed.

We agree that mixing Pearson correlation and Spearman rank correlation was unfortunate, and we now corrected this by removing the use of Pearson correlation from the manuscript. We now also explicitly define R^2 (Line 99).

7. The authors excluded WM115 based on its outlier status in viability predictions using either proteomics or transcriptomics data. Is that a good reason? How many dose points were used for the drug response?

We originally excluded WM115 based on two observations: firstly, inclusion of WM115 led to unstable sensitivity predictions for Cobimetinib when using leave-one-out cross-validation, secondly WM115 was an outlier with respect to PCA analysis of the proteomics dataset. Based on the reviewer's comment we now thoroughly analysed the stability of our results, and found that replacing leave-one-out cross-validation with repeated ($N \approx 100-1000$) 10-fold cross-validation greatly improved the stability of our results, and consequently we do no longer exclude WM115. We gauge the stability of the drug sensitivity predictions by the standard deviation of the correlation coefficient between predicted and measured AUC (after cross-validation) when using different (random) 10-fold partitions. In addition, we investigate the role of contamination by "duplicate" cell lines, originating from the same patient, by redoing the whole analysis without duplicates. This exclusion did not have any major impact on the general conclusions of the manuscript (Line 86-88, and Supplemental Figures S22).

Nine drug doses following an evenly (log)spaced 1:3 serial dilution were used. We have now added a subsection in Methods describing the pharmacological profiling experiments in greater detail (Line 372).

8. There is fairly broad consensus that Elastic net is preferable over pure lasso and this should be used as benchmark.

To address the reviewer's comment, we now benchmark elastic net vs Lasso (a special case of Elastic net), together with several other machine learning algorithms (Supplemental Figure S6+S7), and find that elastic net and Lasso have identical performance in our dataset, however, Lasso has the advantage of being less computationally expensive.

9. line 116: 'When we tested the AKT-inhibitor Ipatasertib (GDC-0068) we found that it inhibited the growth of cell lines in the p-AKT high cluster more effectively than in the PTEN high cluster (adjusted $p = 0.0014$)'. I take it the authors hadn't tested the AKT-inhibitor before? Maybe they could phrase it as: "Motivated by the strong clustering based on components of the PI3K pathway, we tested the AKT-inhibitor x in addition to the y other drugs in the primary screen."

The AKT inhibitor was part of the original drug screen, however, it was filtered out because of its low coefficient of variation in AUC across the cell line panel, as is apparent from Figure 2e. We now state this explicitly in the manuscript (Line 125).

10. line 168: The authors state: 'To reduce the complexity of our PLS models we tested to limit the number of (phospho)proteins to 10'. Sentence does not read well and why was 10 selected?

We removed this part from the manuscript. Instead we now show how cross-validated RMS error depends on the number of non-zero Lasso variables for two of the most predictive/interesting drugs, MMAE and Vemurafenib (Supplemental Figure S8-S11). The smaller the number of non-zero variables, the easier it is to implement the “composite biomarker” in practice.

11. line 225: 'Drugs with low variance ($\sigma/\mu = < 0.135$)'. This is not variance but coefficient of variation.

We have changed the wording accordingly throughout the manuscript.

12. line 231: 'Furthermore, sound effect sizes ($\rho = 0.5$) were deemed insignificant due to small sample size.' ρ is not an effect size. #

We have changed the wording accordingly.

13. Throughout: It is princiPAL component analysis, not princiPLE component analysis.

We corrected this.

14. Supp fig S3: Please denote what red and blue are. I assume it's the coefficient from the PLS?

All supplementary figures now have axes labels and colorbar labels.

15. Supp figs S4-S5: What are linear fits in this context? And why are we looking at Pearson correlation here instead of Spearman?

The Supplemental Figures section has been completely overhauled, and these figures are no longer part of the manuscript.

16. Supp figs S7-S10: The lasso plots are interesting, but not really discussed anywhere. If they're not discussed anywhere, I'd probably leave them out.

We show only a few examples of Lasso plots in the Supplement Figures, which we refer to from the main text (Line 187, Supplemental Figure S8-S11). The fact that PLS and Lasso agree on the most predictive features is an important non-trivial test showing that the drug sensitivity predications are not due to artefacts.

17. Supp figs S12-13: What's the red color here? Pearson correlation between measured and predicted viability?

See comment 14.

18. The following figures really look alike and it takes some effort (going back to the text, reading the legend) to see where they are different. A small addition to the figure would take this effort away.

See comment 14+15.

19. Supp figs S15-16: Indicate in the figure that this is per-cancer, and indicate that one is CCLE and the other one is CTRP.

See comment 14+15.

20. Supp figs S17-19: Clearly indicate that this is a pan-cancer analysis (it says pan-cancer data in the legend, but you can still do a per-cancer analysis on a pan-cancer data set) on CTRP (to distinguish it from 5a-c, where you should probably indicate that it is a pan-cancer and CCLE analysis).

See comment 14+15.

Reviewer #1 Review

Report for Author:

The following broad claims were made in the previous version of the manuscript:

1. that proteomic state is more predictive of drug sensitivity than transcriptomic state;
2. that tissue type is equally predictive of response in the pan cancer setting as molecular data
3. that a proteomics predictor trained on one cancer type (melanoma) can not be transferred to another cancer type (endometrial) and that
4. that on-drug proteomics measurements are not more predictive for response than data collected in the treatment naive setting for a specific drug.

After carrying out the reviewer comments, the authors managed to disprove all but one (Claim 3) of the claims made in the previous version of the manuscript. While Claim 3 still holds in the sense that for only 5 of the 25 drugs a predictor can be transferred from melanoma to endometrium, the authors also show that predictors trained on all-but-one cancer type can (in most cases) be transferred to the left out cancer type. Clearly the training set size plays an important role here, as the authors demonstrate, so there is still more exploration to be done regarding transferability. Here is the list of conclusions in the revised manuscript, as stated in the abstract:

1. drug sensitivity models trained on transcriptomic or proteomic data
2. outperform genomic based models for most drugs;
3. drug sensitivity models can be transferred between tumor types;
4. transcriptomic/proteomic signals may be alternative biomarker candidates for the stratification of patients without known genomic markers.

Unfortunately these claims/conclusions are not novel as 1 and 2 have both been demonstrated in Iorio et al, 2016 (except for the proteomic markers, which have been investigated in later publications) and 3 is a consequence of 1.

Taken together, the authors did a reasonable effort to address the comments. Unfortunately, as a consequence, the new results contained in the revised manuscript are not sufficiently novel to warrant publication in this journal. Below we only treat the major comments.

Major comments

1. While the study looks at potentially interesting problems fairly strong claims are made as outlined above and these can not be fully supported by the data presented. Here are some specifics:

1.1 The claim that proteomics is more predictive than mRNA can, so some degree be made based on the data in Panel 4b as there are more red cells (significant) in the proteomics column as in the mRNA column. However, there are only 27 drugs so adding a significance to this number will likely be hard. Second, each data type is predictive for a different set of drugs - admittedly there is also a large amount of overlap. This is a more accurate and more important message. It is quite unexpected that mRNA data is not predictive of Vemurafenib response. In addition, one might question the validity of such a comparison on melanoma which is so strongly driven by MAPK signaling. Finally, showing this in only melanoma is interesting, but the message would be far more relevant and strong if is demonstrated in at least two additional cancer types (where the deck is not so obviously stacked in favor of phospho-proteomics).

Author response: To address the reviewer's comment we now compare genomic/transcriptomic/proteomic data as predictor of drug sensitivity in eight additional

tumor types using publicly available CCLE data, showing that transcriptomic and proteomic data yield more accurate drug sensitivity predictions than genomic data for most drugs, similar to our in-house melanoma panel results (see Figure 5). In addition, we now interpret our results strictly conservatively, which is also reflected in the updated title of the manuscript. We agree that there is a substantial overlap between transcriptomic and proteomic drug sensitivity predictions. We now correlate the proteomic and transcriptomic datasets, see answer to Comment 2 below.

Reviewer reply: The authors carried out the suggested experiments and these disprove their initial point. Now the conclusion is that proteomics and transcriptomics are equally predictive, which has also been demonstrated in, amongst other, Roumeliotis et al. Cell Reports, 2017.

1.2 The authors show, for a single drug (Dacomitinib), that the pre-treatment data is roughly equally predictive than the on-treatment data. Once again, this is an important hypothesis to investigate, but it is hardly convincing to show this for a single drug. This should be performed far more extensively to draw any meaningful conclusions. This is the type of result that gets cited out of context very easily. In this context, the LINCS project contains a far more exhaustive data set. In addition, while the performance may be similar, it would be interesting to know whether the posttreatment measurements show different phenomena than the pre-treatment data. For example, are the residuals correlated?

Author response: We agree that using one single drug is not enough to generalise the idea that post-treatment data is no more predictive than pre-treatment data. Additionally, when preparing this resubmission we became aware of that the Cobimetinib AUC predictions were sensitive to small perturbations in the input data, such as small changes in filtering settings or excluding individual cell lines from the analysis (see also answer to Comment 7). Since we are no longer confident in the results presented in this section, we have decided to completely remove it from the manuscript. While the LINCS project has amazing data for a limited number of cell lines, our analysis requires data for a large number of cell lines of the same tumor type with matched drug sensitivity. We therefore believe that LINCS, in its current state, would not be the optimal route to compare post-treatment vs pre-treatment data. We would like to point out that upon submission, our proteomic dataset was, to the best of our knowledge, the largest available dataset for a cell line panel of a single tumor type with matched drug sensitivity data.

Reviewer reply: Not applicable anymore as authors discovered that the data are not robust enough to include in the manuscript. Regarding the novelty of the dataset: the mass spectrometry dataset published by Roumeliotis et al, Cell Report, 2107 is of comparable size in terms of the number of samples (n=50) but larger in terms of the number of features (thousands of phosphosites and proteins).

1.3 The comparison of the classifier on melanoma and endometrial cancer and the associated conclusion that classifiers can not be transferred from one cancer type to the next is of little value as it is a single case. Again, it is interesting to know, but based on these results one can hardly make a statement such as 'This confirms that signaling states are predictive of drug response, but that these, in general, are not conserved across tumor types'.

Author response: We agree that generalising this point requires additional data. To address the reviewer's comment we now investigate the transferability of drug sensitivity models for eight additional tumor types using both proteomic and transcriptomic data from CCLE (Figure 5e).

Reviewer reply: We appreciate the effort to thoroughly investigate this point. As only 5 of the 25 drug predictors can be transferred from melanoma to endometrial cancer we agree with the authors that only a 'certain degree cross-cancer predictability is possible'. In the cross transfer between all-but-one cancer type, there seems to be better transferability of a proteomic and transcriptomic classifier trained on all-but-one cancer type and applied to the left-out cancer type. This implies that there are pan-cancer biomarkers on the transcriptomic and proteomic level. A very similar analysis, albeit not in this leave-one-cancer-type out format and only for transcriptomics data, this has already been demonstrated in Iorio et al, 2016, Figure 5. This is, in my opinion, the only remaining novel finding in the manuscript, but is as such not sufficient for publication in this journal.

2. The authors present a new data type (phospho-proteomics, albeit limited in size). It will be very useful to relate this data type to the existing data types (mutations and RNAseq). For example, chart the associations between mutation and RNAseq data and phospho-proteomics. Which variables are correlated? What is the uniqueness and redundancy between the different data types? This is important to know as this provides a better stage for comparing the predictive capacity for drug response.

Author response: To address the reviewer's comment, we now correlate the proteomic and transcriptomic datasets. More precisely, we use PLS to predict proteomic markers from transcriptomic data, and find that the majority of proteomic markers can be rather well predicted from transcriptomic data (Supplemental Figure S15). This could explain why there is a substantial overlap in predictability of drug sensitivity from either the proteomic or transcriptomic dataset.

Reviewer Reply: For transcriptomics and proteomics, this is the expected outcome: there is a large degree of collinearity between transcriptomics and proteomics, which explains their comparable overall performance. However, we also suggested that the correlation between mutations and proteomics/transcriptomics be investigated to learn which drugs can be predicted by mutations and not by the other data types and vice versa. The author response only partially addresses our comment.

3. In the comparison of the mRNA and proteomics there are a number of differences that could explain the differences in performance. One of these is dimensionality. The authors do perform a selection of genes in specific pathways, but reducing the number of features in the mRNA expression dataset based on the variance in the features is a simple but necessary experiment. Since these data types will likely reveal the same biological phenomena employing different genes (MAPK activity is reflected by phosphorylation events in the pathway, while RNAseq will capture downstream expression programs) selecting the same genes/proteins is also an interesting approach but likely not the optimal one.

Author response: We fully agree that feature selection is a crucial part of machine learning. In the revised manuscript, we now systematically assess various strategies for reducing high-dimensional transcriptomic data before making drug sensitivity predictions, both using our in-house melanoma panel and CCLE data (see Supplemental Figures S12+S21). An interesting outcome is that the PLS drug sensitivity predictions are rather insensitive to the precise selection of genes, presumably due to correlations between groups of genes.

Reviewer reply: This is of academic interest as the new results (Comment 1) now show that there is no performance difference between proteomics and transcriptomics. The authors did show that proteomics and transcriptomics data show a large degree of correlation.

4. The authors do perform integration of RNAseq and proteomics data, but in the most simplistic way possible - early integration. It would be interesting to explore a number of interesting alternatives, including intermediate or late integration. In this way the complementarity of the data types will become more evident. Reliable biomarkers will likely emerge from both data types.

Author response: To address the reviewer's comment, we now perform also late integration, where the proteomic and transcriptomic models are weighted together by their respective inverse RMS error after cross-validation. Late and early integration resulted in similar performance (Supplemental Figure S13).

Reviewer reply: great that the authors performed this experiment, and that they have demonstrated that there is actually no benefit in combining the data - which is counter-intuitive and in contrast to what has been demonstrated earlier. However, the goal was to explore an interesting problem: determine where the data types are complementary regarding drug prediction, by, for example employing deflation approaches such as those proposed by De Bin et al Stat Med. 2014 Dec 30;33(30):5310-29. Unfortunately, apart from the BRAF+Vemurafenib case that seems to be unique to mutation data, this was not fully explored.

5. The authors use 'mean viability' as measure of drug response. If the drug concentrations are equally spaced, this is the same as the Area Under the dose response Curve (AUC). If this is the case, the authors should rather refer to it as the AUC. If the concentrations are not equidistant, we suggest calculating the actual AUC and using this. Otherwise results should be repeated with the IC50.

Author response: The drug concentrations are equally spaced in log-space, hence we now follow the reviewer's advice and replace "mean viability" with "AUC" throughout the manuscript. We have also added a subsection in Methods describing the drug sensitivity screening experiments in greater detail (Line 372).

Reviewer #2 Review

Report for Author:

The reviewed manuscript has addressed many but not all of the issues raised by the reviewers. Hence the manuscript has improved but still falls short of the expectations for a paper in this journal.

Specifically,

1. This reviewer feels that far too much of the manuscript is devoted to the almost trivial case of BRAF. Not only is this well known, the authors make several statements in this section that are simply incorrect or confusing. The reason why only some melanoma lines respond to BRAF inhibitors is because they are driven by the V600E mutation. Therefore, it is entirely clear that this genomic information predicts drug response. It is therefore trivial to conclude that "Our results show that the key driver mutation in melanoma, BRAF V600E/K, was only a strong predictor of drug sensitivity for drugs that target the mutated molecule itself." Similarly, the title of the next section states an entirely trivial fact for the same reason. The V600E is what is driving the tumor. Not the other mutations in exons.

In this regard it is also confusing that the authors state in the introduction that "This often renders cross-entity biomarkers too unspecific to be used to stratify patients". If a melanoma patient presents with a V600E mutation in BRAF, the patient will almost certainly be treated with a BRAF inhibitor.

Next, the statement "Again, only the BRAF V600E/K dependent drugs Vemurafenib and Dabrafenib could be predicted significantly..." is plain wrong. Both drugs inhibit BRAF wt just as much as the V600E mutant. It is a common misconception that these BRAF inhibitors are specific to the mutant protein. They are not. The only reason why these drugs work in BRAF V600E patients is because the mutant protein drives the tumor.

Another main criticism to the initial manuscript was that none of the new hypothesis was validated by independent experiments. This was not addressed in the revision.

June 4, 2019

RE: Life Science Alliance Manuscript #LSA-2019-00445-T

Prof Nils Blüthgen
Charite Universitätsmedizin Berlin
Institute of Pathology and IRI for the Life Sciences
Chariteplatz 1
Berlin D-10115
Germany

Dear Dr. Blüthgen,

Thank you for transferring your revised manuscript entitled "Signaling and expression states of cancer cells predict sensitivity to targeted and cytotoxic drugs" to Life Science Alliance.

Your work was reviewed at another journal twice before, and the editors transferred those reviewer reports and your response to us with your permission. The reviewers appreciated the technical quality of your work and the thorough revision performed, but thought that the conceptual advance provided remained somewhat limited. This is not a concern for us, and we would thus be happy to publish your paper in Life Science Alliance pending final revisions necessary to meet our formatting guidelines:

- please provide a final point-by-point response to the remaining concerns of the reviewers, including references for drug specificity towards BRAFV600E/K
- please upload all main figures as individual files, the legends should remain in the manuscript word docx file
- please add a callout to Fig 2E in the manuscript text
- please provide information in the Data availability section on NGS data accession codes
- I would like to suggest to change the title slightly. How about: "The cancer cell proteome and transcriptome predicts sensitivity to targeted and cytotoxic drugs"

A. FINAL FILES:

B. MANUSCRIPT ORGANIZATION AND FORMATTING:

Sincerely,

Andrea Leibfried, PhD
Executive Editor
Life Science Alliance

Meyerhofstr. 1
69117 Heidelberg, Germany
t +49 6221 8891 502
e a.leibfried@life-science-alliance.org
www.life-science-alliance.org

Major comments

1. While the study looks at potentially interesting problems fairly strong claims are made as outlined above and these can not be fully supported by the data presented. Here are some specifics:

1.1 The claim that proteomics is more predictive than mRNA can, so some degree be made based on the data in Panel 4b as there are more red cells (significant) in the proteomics column as in the mRNA column. However, there are only 27 drugs so adding a significance to this number will likely be hard. Second, each data type is predictive for a different set of drugs - admittedly there is also a large amount of overlap. This is a more accurate and more important message. It is quite unexpected that mRNA data is not predictive of Vemurafenib response. In addition, one might question the validity of such a comparison on melanoma which is so strongly driven by MAPK signaling. Finally, showing this in only melanoma is interesting, but the message would be far more relevant and strong if it is demonstrated in at least two additional cancer types (where the deck is not so obviously stacked in favor of phospho-proteomics).

Author response: To address the reviewer's comment we now compare genomic/transcriptomic/proteomic data as predictor of drug sensitivity in eight additional tumor types using publicly available CCLE data, showing that transcriptomic and proteomic data yield more accurate drug sensitivity predictions than genomic data for most drugs, similar to our in-house melanoma panel results (see Figure 5). In addition, we now interpret our results strictly conservatively, which is also reflected in the updated title of the manuscript. We agree that there is a substantial overlap between transcriptomic and proteomic drug sensitivity predictions. We now correlate the proteomic and transcriptomic datasets, see answer to Comment 2 below.

Reviewer reply: The authors carried out the suggested experiments and these disprove their initial point. Now the conclusion is that proteomics and transcriptomics are equally predictive, which has also been demonstrated in, amongst other, Roumeliotis et al. Cell Reports, 2017.

We agree with the reviewer that this is consistent with Roumeliotis et al.

1.2 The authors show, for a single drug (Dacomitinib), that the pre-treatment data is roughly equally predictive than the on-treatment data. Once again, this is an important hypothesis to investigate, but it is hardly convincing to show this for a single drug. This should be performed far more extensively to draw any meaningful conclusions. This is the type of result that gets cited out of context very easily. In this context, the LINCS project contains a far more exhaustive data set. In addition, while the performance may be similar, it would be interesting to know whether the posttreatment measurements show different phenomena than the pre-treatment data. For example, are the residuals correlated?

Author response: We agree that using one single drug is not enough to generalise the idea that post-treatment data is no more predictive than pre-treatment data. Additionally, when preparing this resubmission we became aware of that the Cobimetinib AUC predictions were sensitive to small perturbations in the input data, such as small changes in filtering settings or excluding individual cell lines from the analysis (see also answer to Comment 7). Since we are no longer confident in the results presented in this section, we have decided to completely remove it from the manuscript. While the LINCS project has amazing data for a limited number of cell lines, our analysis requires data for a large number of cell lines of the same tumor type with matched drug sensitivity. We therefore believe that LINCS, in its current state, would not be the optimal route to compare post-treatment vs pre-treatment data. We would like to point out that upon submission, our proteomic dataset was, to the best of our knowledge, the largest available dataset for a cell line panel of a single tumor type with matched drug sensitivity data.

Reviewer reply: Not applicable anymore as authors discovered that the data are not robust enough to include in the manuscript. Regarding the novelty of the dataset: the mass spectrometry dataset published by Roumeliotis et al, Cell Report, 2107 is of comparable size in terms of the number of samples (n=50) but larger in terms of the number of features (thousands of phosphosites and proteins).

We agree with the reviewer.

1.3 The comparison of the classifier on melanoma and endometrial cancer and the associated conclusion that classifiers can not be transferred from one cancer type to the next is of little value as it is a single case. Again, it is interesting to know, but based on these results one can hardly make a statement such as 'This confirms that signaling states are predictive of drug response, but that these, in general, are not conserved across tumor types'.

Author response: We agree that generalising this point requires additional data. To address the reviewer's comment we now investigate the transferability of drug sensitivity models for eight additional tumor types using both proteomic and transcriptomic data from CCLE (Figure 5e).

Reviewer reply: We appreciate the effort to thoroughly investigate this point. As only 5 of the 25 drug predictors can be transferred from melanoma to endometrial cancer we agree with the authors that only a 'certain degree cross-cancer predictability is possible'. In the cross transfer between all-but-one cancer type, there seems to be better transferability of a proteomic and transcriptomic classifier trained on all-but-one cancer type and applied to the left-out cancer type. This implies that there are pan-cancer biomarkers on the transcriptomic and proteomic level. A very similar analysis, albeit not in this leave-one-cancer-type out format and only for transcriptomics data, this has already been demonstrated in Iorio et al, 2016, Figure 5. This is, in my opinion, the only remaining novel finding in the manuscript, but is as such not sufficient for publication in this journal.

We agree with the reviewer.

2. The authors present a new data type (phospho-proteomics, albeit limited in size). It will be very useful to relate this data type to the existing data types (mutations and RNAseq). For example, chart the associations between mutation and RNAseq data and phospho-proteomics. Which variables are correlated? What is the uniqueness and redundancy between the different data types? This is important to know as this provides a better stage for comparing the predictive capacity for drug response.

Author response: To address the reviewer's comment, we now correlate the proteomic and transcriptomic datasets. More precisely, we use PLS to predict proteomic markers from transcriptomic data, and find that the majority of proteomic markers can be rather well predicted from transcriptomic data (Supplemental Figure S15). This could explain why there is a substantial overlap in predictability of drug sensitivity from either the proteomic or transcriptomic dataset.

Reviewer Reply: For transcriptomics and proteomics, this is the expected outcome: there is a large degree of collinearity between transcriptomics and proteomics, which explains their comparable overall performance. However, we also suggested that the correlation between mutations and proteomics/transcriptomics be investigated to learn which drugs can be predicted by mutations and not by the other data types and vice versa. The author response only partially addresses our comment.

We did the analysis, but observed a very low predictability of proteomic states from genomic data. We have now included this in the supplement (supplementary figure S16)

3. In the comparison of the mRNA and proteomics there are a number of differences that could explain the differences in performance. One of these is dimensionality. The authors do perform a selection of genes in specific pathways, but reducing the number of features in the mRNA expression dataset based on the variance in the features is a simple but necessary experiment. Since these data types will likely reveal the same biological phenomena employing different genes (MAPK activity is reflected by phosphorylation events in the pathway, while RNAseq will capture downstream expression programs) selecting the same genes/proteins is also an interesting approach but likely not the optimal one.

Author response: We fully agree that feature selection is a crucial part of machine learning. In the revised manuscript, we now systematically assess various strategies for reducing high-dimensional transcriptomic data before making drug sensitivity predictions, both using our in-house melanoma panel and CCLE data (see Supplemental Figures S12+S21). An interesting outcome is that the PLS drug sensitivity predictions are rather insensitive to the precise selection of genes, presumably due to correlations between groups of genes.

Reviewer reply: This is of academic interest as the new results (Comment 1) now show that there is no performance difference between proteomics and transcriptomics. The authors did show that proteomics and transcriptomics data show a large degree of correlation.

We agree with the reviewer.

4. The authors do perform integration of RNAseq and proteomics data, but in the most simplistic way possible - early integration. It would be interesting to explore a number of interesting alternatives, including intermediate or late integration. In this way the complementarity of the data types will become more evident. Reliable biomarkers will likely emerge from both data types.

Author response: To address the reviewer's comment, we now perform also late integration, where the proteomic and transcriptomic models are weighted together by their respective inverse RMS error after cross-validation. Late and early integration resulted in similar performance (Supplemental Figure S13).

Reviewer reply: great that the authors performed this experiment, and that they have demonstrated that there is actually no benefit in combining the data - which is counter-intuitive and in contrast to what has been demonstrated earlier. However, the goal was to explore an interesting problem: determine where the data types are complementary regarding drug prediction, by, for example employing deflation approaches such as those proposed by De Bin et al Stat Med. 2014 Dec 30;33(30):5310-29. Unfortunately, apart from the BRAF+Vemurafenib case that seems to be unique to mutation data, this was not fully explored.

We thank the reviewer, but feel that exploring further integration approaches (like deflation) is beyond the scope of this manuscript.

5. The authors use 'mean viability' as measure of drug response. If the drug concentrations are equally spaced, this is the same as the Area Under the dose response Curve (AUC). If this is the case, the authors should rather refer to it as the AUC. If the concentrations are not equidistant, we suggest calculating the actual AUC and using this. Otherwise results should be repeated with the IC50.

Author response: The drug concentrations are equally spaced in log-space, hence we now follow the reviewer's advice and replace "mean viability" with "AUC" throughout the manuscript. We have also added a subsection in Methods describing the drug sensitivity screening experiments in greater detail (Line 372).

Reviewer Reply: OK

Reviewer #2:

The reviewed manuscript has addressed many but not all of the issues raised by the reviewers. Hence the manuscript has improved but still falls short of the expectations for a paper in this journal. Specifically,

1. This reviewer feels that far too much of the manuscript is devoted to the almost trivial case of BRAF. Not only is this well known, the authors make several statements in this section that are simply incorrect or confusing. The reason why only some melanoma lines respond to BRAF inhibitors is because they are driven by the V600E mutation. Therefore, it is entirely clear that this genomic information predicts drug response. It is therefore trivial to conclude that "Our results show that the key driver mutation in melanoma, BRAF V600E/K, was only a strong predictor of drug sensitivity for drugs that target the mutated molecule itself."

Similarly, the title of the next section states an entirely trivial fact for the same reason. The V600E is what is driving the tumor. Not the other mutations in exons.

In this regard it is also confusing that the authors state in the introduction that "This often renders cross-entity biomarkers too unspecific to be used to stratify patients". If a melanoma patient presents with a V600E mutation in BRAF, the patient will almost certainly be treated with a BRAF inhibitor.

Next, the statement "Again, only the BRAF V600E/K dependent drugs Vemurafenib and Dabrafenib could be predicted significantly..." is plain wrong. Both drugs inhibit BRAF wt just as much as the V600E mutant. It is a common misconception that these BRAF inhibitors are specific to the mutant protein. They are not. The only reason why these drugs work in BRAF V600E patients is because the mutant protein drives the tumor.

We disagree with the statements by this reviewer, most importantly with the statement "Both drugs inhibit BRAF wt just as much as the V600E mutant". Biochemically, this is not true:

- Vemurafenib has a 3-fold lower IC₅₀ for BRAF V600E compared to BRAF wild type (Bollag et al, 2010, Nature, 467(7315):596-9, supplementary table 1).
- Dabrafenib has a 7-fold lower IC₅₀ for the mutant protein (Rheault et al, 2013, ACS Med Chem Lett., supplementary table S1).

In addition, these BRAF inhibitors cause paradoxical activation, and by that tend to activate MAPK signaling in BRAF WT cells.

While we agree that many or most BRAF mutant melanoma cell lines might be driven by this oncogene, it is not a priori clear that other melanoma cell lines do not rely on RAF signaling. Most importantly, we see that the response to MEK inhibitors is not well explainable by the BRAF V600E mutation and does not correlate well with the response to BRAF inhibitors. Therefore, we remain confident that our statements are correct and in line with biochemical data.

Another main criticism to the initial manuscript was that none of the new hypothesis was validated by independent experiments. This was not addressed in the revision.

We also don't agree. We validated our main results on a large published cell line panel.

June 14, 2019

RE: Life Science Alliance Manuscript #LSA-2019-00445-TR

Prof. Nils Blüthgen
Charite Universitätsmedizin Berlin
Institute of Pathology and IRI for the Life Sciences
Chariteplatz 1
Berlin D-10115
Germany

Dear Dr. Blüthgen,

Thank you for submitting your Research Article entitled "The cancer cell proteome and transcriptome predicts sensitivity to targeted and cytotoxic drugs". It is a pleasure to let you know that your manuscript is now accepted for publication in Life Science Alliance. Congratulations on this interesting work.

DISTRIBUTION OF MATERIALS:

Again, congratulations on a very nice paper. I hope you found the review process to be constructive and are pleased with how the manuscript was handled editorially. We look forward to future exciting submissions from your lab.

Sincerely,

Andrea Leibfried, PhD
Executive Editor
Life Science Alliance
Meyershofstr. 1
69117 Heidelberg, Germany
t +49 6221 8891 502
e a.leibfried@life-science-alliance.org
www.life-science-alliance.org